# Measures of distortion for machine learning

**Leena Chennuru Vankadara**
University of Tübingen
Max Planck Institute for Intelligent Systems, Tübingen
`leena.chennuru@tuebingen.mpg.de`

**Ulrike von Luxburg**
University of Tübingen
Max Planck Institute for Intelligent Systems, Tübingen
`luxburg@informatik.uni-tuebingen.de`

## Abstract

Given data from a general metric space, one of the standard machine learning pipelines is to first embed the data into a Euclidean space and subsequently apply machine learning algorithms to analyze the data. The quality of such an embedding is typically described in terms of a distortion measure. In this paper, we show that many of the existing distortion measures behave in an undesired way, when considered from a machine learning point of view. We investigate desirable properties of distortion measures and formally prove that most of the existing measures fail to satisfy these properties. These theoretical findings are supported by simulations, which for example demonstrate that existing distortion measures are not robust to noise or outliers and cannot serve as good indicators for classification accuracy. As an alternative, we suggest a new measure of distortion, called $\sigma$-distortion. We can show both in theory and in experiments that it satisfies all desirable properties and is a better candidate to evaluate distortion in the context of machine learning.

## 1 Introduction

Given data from a general metric space, one of the standard machine learning pipelines is to first embed the data into a Euclidean space (for example using an unsupervised algorithm such as Isomap, locally linear embedding, maximum variance unfolding, etc) and subsequently apply out of the box machine learning algorithms to analyze the data. Typically, the quality of such an embedding is described in terms of a distortion measure that summarizes how the distances between the embedded points deviate from the original distances. Many distortion measures have been used in the past, the most prominent ones being worst case distortion, $l_q$-distortion (Abraham, Bartal, and Neiman, 2011), average distortion(Abraham, Bartal, and Neiman, 2011), $\epsilon$-distortion (Abraham, Bartal, Kleinberg, et al., 2005), k-local distortion (Abraham, Bartal, and Neiman, 2007) and scaling distortion (Abraham, Bartal, Kleinberg, et al., 2005). Such distortion measures are sometimes evaluated in hindsight to evaluate the quality of an embedding, and sometimes used directly as objective functions in embedding algorithms, for example the stress functions that are commonly used in different variants of Multidimensional scaling (Cox and Cox, 2000). There also exist embedding algorithms with completely different objectives. For instance, t-SNE (Maaten and Hinton, 2008) employs an objective function that aims to enhance the cluster structure present in the data. In this paper, however, we restrict our analysis to distortion measures that evaluate the quality of distance preserving embeddings.

From a theoretical computer science point of view, many aspects of distance preservation of emeddings are well understood. For example, Bourgain's theorem (Bourgain, 1985) and the Johnson–Lindenstrauss Lemma (Johnson and Lindenstrauss, 1984) state that any finite metric space

of $n$ points can be embedded into a Euclidean space of dimension $\mathcal{O}(\log n)$ with worst case distortion $\mathcal{O}(\log n)$. Many related results exist (Gupta, Krauthgamer, and Lee, 2003; Abraham, Bartal, and Neiman, 2008; Abraham, Bartal, and Neiman, 2011; Abraham, Bartal, Kleinberg, et al., 2005; Abraham, Bartal, and Neiman, 2007; Abraham, Bartal, and Neiman, 2011; Semmes, 1996).

However, from a machine learning point of view, these results are not entirely satisfactory. The typical distortion guarantees from theoretical computer science focus on a finite metric space. However, in machine learning, we are ultimately interested in consistency statements: given a sample of $n$ points from some underlying space, we would like to measure the distortion of an embedding algorithm as $n \to \infty$. In particular, the dimension of the embedding space should be constant and not grow with $n$, because we want to relate the geometry of the original underlying space to the geometry of the embedding space. Hence, many of the guarantees that are nice from a theoretical computer science point of view (for example, because they provide approximation guarantees for NP hard problems) miss the point when applied to machine learning (either in theory or in practice, see below).

Ideally, in machine learning we would like to use the distortion measure as an indication of the quality of an embedding. We would hope that when we compare several embeddings, choosing the one with smaller distortion would lead to better machine learning results (at least in tendency). However, when we empirically investigated the behavior of existing distortion measures, we were surprised to see that they behave quite erratically and often do not serve this purpose at all (see Section 4).

In pursuit of a more meaningful measure of distortion in the context of machine learning, we take a systematic approach in this paper. We identify a set of properties that are essential for any distortion measure. In light of these properties, we propose a new measure of distortion that is designed towards machine learning applications: the $\sigma$-distortion. We prove in theory and through simulations that our new measure of distortion satisfies many of the properties that are important for machine learning, while all the other measures of distortion have serious drawbacks and fail to satisfy all of the properties. These results can be summarized in the following table (where each column corresponds to one measure of distortion and each row to one desirable property, see Section 2 for notation and definitions):

| Property/Distortion measure | $\sigma$(sigma) | wc | avg($l_q$) | navg | k-local | $\epsilon$(epsilon) |
|:---:|:---:|:---:|:---:|:---:|:---:|:---:|
| Translation invariance | ✓ | ✓ | ✓ | ✓ | ✓ | ✓ |
| Monotonicity | ✓ | ✓ | ✗ | ✓ | ✓ | ✓ |
| Scale invariance | ✓ | ✓ | ✗ | ✓ | ✓ | ✓ |
| Robustness to outliers | ✓ | ✗ | ✓ | ✗ | ✗ | ✓ |
| Robustness to noise | ✓ | ✗ | ✗ | ✗ | ✗ | ✓ |
| Incorporation of probability | ✓ | ✗ | ✗ | ✗ | ✗ | ✗ |
| Constant distortion embeddings | ✓ | ✗ | ✓ | ? | ✓ | ✓ |

## 2   Existing measures of distortion

Let $(X, d_X)$ and $(Y, d_Y)$ be arbitrary finite metric spaces. Let $\binom{X}{2} := \{\{u, v\} \mid u, v \in X, \ u \neq v\}$ and for any $n \in I\!N$, let $[n]$ denote the set $\{1, 2, ..., n\}$. An embedding of $X$ into $Y$ is an injective mapping $f : (X, d_X) \to (Y, d_Y)$. Let $P$ be a probability distribution on $X$, and $\Pi := P \times P$ the product distribution on $X \times X$. Distortion measures aim to quantify the deviation of an embedding from isometry. Intuitively, the distortion of such an embedding is supposed to measure how far the new distances $d_Y(f(u), f(v))$ between the embedded points deviate from the original distances $d_X(u, v)$. Virtually all the existing distortion measures are summary statistics of the pairwise ratios

$$\rho_f(u, v) = d_Y(f(u), f(v))/d_X(u, v)$$

with $u, v \in X$. The intention is to capture the property that if the ratios $d_Y(f(u), f(v))/d_X(u, v)$ are close to 1 for many pairs of points $u, v$, then the distortion is small. The most popular measures

of distortion are the following ones:

**Worst case distortion:** $\Phi_{wc}(f) := \left( \max\limits_{(u,v) \in \binom{X}{2}} \rho_f(u,v) \right) \cdot \left( \max\limits_{(u,v) \in \binom{X}{2}} \dfrac{1}{\rho_f(u,v)} \right).$

**Average case distortion:** $\Phi_{avg}(f) := \dfrac{2}{n(n-1)} \sum\limits_{(u,v) \in \binom{X}{2}} \rho_f(u,v).$

**Normalized avg distortion:** $\Phi_{navg}(f) := \dfrac{2}{n(n-1)} \sum\limits_{u \neq v \in X} \dfrac{\rho_f(u,v)}{\alpha}$ with $\alpha = \min\limits_{u \neq v \in X} \rho_f(u,v).$

$l_q$**-distortion** (with $1 \leq q < \infty$): $\Phi_{l_q}(f) := \mathbb{E}_\Pi(\rho_f(u,v)^q)^{\frac{1}{q}}.$

$\epsilon$**-distortion** ($\forall\, 0 < \epsilon < 1$): $\Phi_\epsilon(f) := \min\limits_{S \subset \binom{X}{2}, |S| \geq (1-\epsilon)\binom{n}{2}} \Phi_{wc}(f_S)$, where $f_S$ denotes the restriction of $f$ to $S$.

**k-local distortion:** $\Phi_{klocal}(f) := \left( \max\limits_{u \in X, v \in \text{kNN}(u), u \neq v} \rho_f(u,v) \right) \cdot \left( \max\limits_{u \in X, v \in \text{kNN}(u) u \neq v} \dfrac{1}{\rho_f(u,v)} \right),$

where $\text{kNN}(u)$ denotes the set of $k$ nearest neighbours of $u$.

The different measures of distortion put their focus on different aspects: the worst case among all pairs of points ($\Phi_{wc}$), the worst case excluding pathological outliers ($\Phi_\epsilon$), the average case ($\Phi_{avg}, \Phi_{navg}, \Phi_{l_q}$) or distortions that are just evaluated between neighboring points ($\Phi_{klocal}$).

From a conceptual level, all these measures of distortion make sense, and it is not obvious why one should prefer one over the other. However, when we studied them in practice, we found different sources of undesired behavior for many of them. For example, many of them behave in a quite unstable or even erratic manner, due to high sensitivity to outliers or because they are not invariant with respect to rescaling. To study these issues more systematically, we will now identify a set of properties that any measure of distortion should satisfy in the context of machine learning applications. In Section 3.2 we then prove which of the existing measures satisfies which properties and find that each of them has particular deficiencies. In Section 3.3 we then introduce a new measure of distortion that does not suffer from these issues, and demonstrate its practical behavior in Section 4.

## 3 Properties of distortion measures

In this section we identify properties that a distortion measure is expected to satisfy in the context of machine learning. In addition to basic properties such as invariance to rescaling and translation, the most important properties should resonate with an appropriate characterization of the quality of an embedding. In the following, let $(X, d_X)$ be an arbitrary metric space, let $Y$ be an arbitrary vector space and let $d_Y$ be a homogeneous and translation invariant metric on $Y$ (See the supplement for the formal definitions). Let $f, g : (X, d_X) \rightarrow (Y, d_Y)$ be two embeddings and let $\Phi$ be any function that is supposed to measure the distortion of any injective mapping from $X$ to $Y$.

### 3.1 Definitions

We start with a set of **basic properties** that should be satisfied by any function that is supposed to provide a measure of distortion, irrespective of the context in which it is applied.

**Scale Invariance** is an essential property for a measure of distortion since embeddings that are merely different in units of measurement (say, kilometers vs centimeters) should not be assigned different values of distortion. Formally, let $f : (X, d_X) \rightarrow (Y, d_Y)$ and $g : (X, d_X) \rightarrow (Y, d_Y)$ be two injective mappings. A distortion measure $\Phi$ is said to be scale invariant if for any $\alpha \in \mathbb{R}$,

$$\forall u \in X, f(u) = \alpha g(u) \implies \Phi(f) = \Phi(g). \tag{1}$$

**Translation Invariance:** A measure of distortion should clearly be invariant to translations: Let $f : (X, d_X) \rightarrow (Y, d_Y)$ and $g : (X, d_X) \rightarrow (Y, d_Y)$ be two injective mappings. A measure of

distortion $\Phi$ is said to be translation invariant if for any $y \in Y$,

$$\forall u \in X, f(u) = g(u) + y; \implies \Phi(f) = \Phi(g). \tag{2}$$

**Monotonicity** captures the property that if distances are preserved more strictly, then the distortion of the corresponding embedding should be smaller. The formal definition is a bit tricky, because one has to be careful about scaling issues. We take care of it by standardizing the embeddings such that the average of the $\rho(u,v)$ is 1. Let $f : (X, d_X) \to (Y, d_Y)$ and $g : (X, d_X) \to (Y, d_Y)$ be embeddings. Define the scaling constants $\alpha(f) = (\frac{2}{n(n-1)}) \sum_{u \neq v \in X} \rho_f(u,v)$ and $\alpha(g) = (\frac{2}{n(n-1)}) \sum_{u \neq v \in X} \rho_g(u,v)$. Then a measure of distortion $\Phi$ is said to be monotonic if

$$\forall u, v \in X, \left( \left( \frac{\rho_f(u,v)}{\alpha(f)} \leq \frac{\rho_g(u,v)}{\alpha(g)} \leq 1 \right) \text{ or } \left( \frac{\rho_f(u,v)}{\alpha(f)} \geq \frac{\rho_g(u,v)}{\alpha(g)} \geq 1 \right) \right) \implies \Phi(f) \geq \Phi(g). \tag{3}$$

After having introduced the basic properties that need to be satisfied such that a function $\Phi$ deserves the term "distortion", we now turn to some **advanced properties** that specifically identify the necessary characteristics of distortion measures in the context of machine learning applications.

**Robustness to outliers:** Outliers are inherent to data processed by machine learning algorithms, and hence a measure of distortion that is too volatile against outliers is not desirable. What we would like to achieve is rather that the influence of a single data point or a single distance value on the measure of distortion is very small. In the spirit of this interpretation, we create two test cases as necessary conditions to deem a measure of distortion robust to outliers.

**Outliers in data**: To verify that the effect of a single data point on the measure of distortion is small, we stipulate that the influence of this point should converge to $0$ as the number $n$ of points goes to infinity. To formulate this property, we compare an isometric embedding to an embedding that is "isometric except for one point". Formally, let $I : (X, d_X) \to (X, d_X)$ be an isometry. Fix arbitrary $x_0, x^* \in X$ and $\beta > 0$. For any $n \in \mathbb{N}$, let $X_n = \{x_1, x_2, ..., x_n\} \subset X \setminus B(x_0, \beta)$. Let $f_n : X_n \cup \{x_0\} \to X$ such that

$$f_n(x) = \begin{cases} x^*, & \text{if } x = x_0. \\ x, & \text{otherwise.} \end{cases} \tag{4}$$

We say that a measure of distortion $\Phi$ is not robust to outliers if $\lim_{n \to \infty} \Phi(f_n) \neq \lim_{n \to \infty} \Phi(I_n)$, where $I_n$ denotes the restriction of the mapping $I$ to $X_n \cup \{x_0\}$. In the formal definition, one needs to make sure that the distortions do not grow arbitrarily fast, which can happen either if points in the original space are too close or if points in the image space are too far from each other. The ball of positive radius $\beta$ prevents the first case, and the fact that we choose $x^*$ as a fixed point prevents the second case.

**Outliers in distances**: To evaluate whether a measure of distortion is robust to outliers in distances, we consider mappings for which at most a constant $(K)$ number of distances are distorted and compare the resulting distortion measure to the one of an isometry. Formally, let $I : (X, d_X) \to (X, d_X)$ be an isometry. Let $X_D = \{x_1, x_2, ...., \} \subset X$. Let $f : X_D \to X$ be an injective mapping such that there exists a constant $K \in \mathbb{N}$ for which the set $G = \{(u,v) \in X_D \times X_D : d_X(f(u), f(v)) \neq d_X(u,v)\}$ satisfies $|G| \leq K$. For any $n \in \mathbb{N}$, let $f_n$ and $I_n$ denote the restriction of the mappings $f$ and $I$, respectively, to $X_n = \{x_1, x_2, ..., x_n\} \subset X_D$. We say that a measure of distortion $\Phi$ is not robust to outliers if $\lim_{n \to \infty} \Phi(f_n) \neq \lim_{n \to \infty} \Phi(I_n)$.

**Incorporation of the probability distribution:** In machine learning, a standard assumption is that the data has been sampled according to some probability distribution from an underlying space. A measure of distortion should be able to take this probability distribution into account, in the sense that distortions of points in high density regions should be more costly than distortions of points in low density regions. We formalize this idea by stipulating that given two different embeddings which are "isometric except for one point", where the two embeddings distort two different points such that the ratios of distorted distances are the same for both the embeddings, then the embedding that distorts the point that occurs with higher probability needs to have a higher value of distortion.

Let $(X, d_X)$ be an arbitrary metric space. Let $X_n = \{x_1, x_2, ..., x_n\}$ be a finite subset of $X$. Let $P$ denote a probability distribution on $X_n$. Fix arbitrary $x^*, y^* \in X_n$ such that $P(x^*) > P(y^*)$.

Let $x', y' \in X$ such that $\forall i \in [n], d_X(x_i, x') = \alpha d_X(x_i, x^*)$ and $d_X(x_i, y') = \alpha d_X(x_i, y^*)$. Let $f, g : X_n \to X$ be two embeddings such that:

$$f(x) = \begin{cases} x', & \text{if } x = x^*. \\ x, & \text{otherwise.} \end{cases} \quad , \quad g(x) = \begin{cases} y', & \text{if } x = y^*. \\ x, & \text{otherwise.} \end{cases}$$

Then a measure of distortion $\Phi$ is said to incorporate the probability distribution $P$ if $\Phi(f) > \Phi(g)$.

**Robustness to noise:** Noisy observations, just as outliers, are common in machine learning. In machine learning applications we would expect that the measure of distortion is smaller if there is less noise on the data. For this property, we do not provide a formal definition. Rather, we conduct experiments to empirically verify whether this is the case in simple settings.

We believe that in order to be useful for machine learning, a measure of distortion should satisfy all the basic as well as the advanced properties. We would like to conclude this list with a last property that is perhaps not absolutely crucial, but **nice to have**: the ability to provide **constant-dimensional embeddings.**

In learning theory we often assume that we are given a set of data points that has been sampled according to some underlying probability distribution, and then we are interested in consistency statements: given a sample of $n$ points, we study the behavior of algorithms as the sample size $n$ goes to infinity. In particular, when we consider embeddings we would hope that as the sample size grows, the geometry of the embedded points "converges" to something that is close to the "true geometry" of the underlying space. In particular, if the underlying space is "simple", we would like to embed into a Euclidean space of constant dimension — the dimension is not supposed to grow with the sample size because we would then have to deal with an infinite-dimensional space in the limit case, which would allow for too complex geometries. For embeddings in a constant-dimensional space, we would then like to bound the distortion, ideally by a quantity that is bounded by a constant that is independent of $n$ and just depends on the geometry of the true underlying space. In general, it is impossible to guarantee the existence of an embedding into Euclidean space with constant dimension and constant distortion (for all the standard measures of distortion, cf. (Semmes, 1999; Semmes, 1996; Abraham, Bartal, and Neiman, 2011)). However, guarantees can be given if we make assumptions on the underlying metric space. For example if $(X, d_X)$ is doubling, it is possible to achieve an embedding into constant dimensional Euclidean space with $\mathcal{O}(1)$ average distortion (but $\Omega(\log n)$ worst case distortion) (Abraham, Bartal, and Neiman, 2011). Hence we stipulate that a measure of distortion that is nice for machine learning should satisfy that if the underlying geometry of the metric space is "nice" (according to an appropriate definition), then we can guarantee that there exists an embedding in **constant dimension with constant distortion.**

### 3.2 Theoretical results: existing distortion measures fail to satisfy all properties

In the following theorem we investigate which measure of distortion satisfies which of the properties mentioned above. All the proofs can be found in the appendix.

**Theorem 1 (Properties of existing distortion measures).** *For all choices of the parameters $1 \leq q < \infty$, $0 < \epsilon < 1$, $1 \leq k \leq n$, the following statements are true:*

*(a) $\Phi_{wc}$, $\Phi_{avg}$, $\Phi_{navg}$, $\Phi_{l_q}$, $\Phi_\epsilon$ and $\Phi_{klocal}$ satisfy the property of **translation invariance**.*

*(b) $\Phi_{wc}$, $\Phi_{navg}$, $\Phi_\epsilon$, $\Phi_{klocal}$ satisfy the properties of **scale invariance** and **monotonicity.** $\Phi_{avg}$ and $\Phi_{l_q}$ fail to satisfy these properties.*

*(c) $\Phi_\epsilon$, $\Phi_{avg}$, $\Phi_{l_q}$ satisfy the property of **robustness to outliers**. The measures $\Phi_{wc}$, $\Phi_{navg}$, $\Phi_{klocal}$ fail to do so.*

*(d) The distortion measures $\Phi_{wc}$, $\Phi_{avg}$, $\Phi_{navg}$, $\Phi_{l_q}$, $\Phi_\epsilon$, $\Phi_{klocal}$ fail to **incorporate a probability distribution** defined on the data space.*

At the current point in time, we do not yet have a formal guarantee towards **robustness to noise**. However, in our experiments we show that $\epsilon-$distortion is considerably robust to noise for certain values of $\epsilon$ and the rest of the distortion measures do not demonstrate robustness to noise.

Regarding the **constant-dimensional embeddings**, there exists a large literature. In the case of average distortion, $\epsilon$-distortion, and k-local distortion for fixed values of $k, \epsilon$, it has been shown

that any finite subset of a doubling metric space (see the supplement for a formal definition) can be embedded into a constant dimensional Euclidean space with $\mathcal{O}(1)$ distortion (Abraham, Bartal, and Neiman, 2011; Abraham, Bartal, and Neiman, 2009). Hence these measures of distortion also satisfy the "nice to have" property. Such a result for Normalized average distortion doesn't exist in the literature to the best of our knowledge. Worstcase distortion, however, fails to satisfy this property (Semmes, 1999; Semmes, 1996).

### 3.3 A new measure of distortion that satisfies all properties

We have seen that all the existing measures of distortion fail to satisfy at least one of the properties that we identified above. In the light of these results, we introduce a **new measure of distortion, the** $\sigma-$**distortion** ($\Phi_\sigma$). The intuition for our definition is as follows. For a given data set $X$, consider a histogram of the ratios $\rho_f(u,v)$. An embedding of high quality should preserve *most* distances as well as possible, that is we would like to see that most of these ratios are close to 1. We characterize this by measuring the "concentration" of the distribution of the ratio of distances ($\rho_f(u,v)$) in terms of the variance. The fact that we consider the variance of this distribution makes our definition robust against outliers (one of the properties which most of the other distortion measures fail to satisfy). By a rescaling step we will achieve invariance with respect to scale. Furthermore we will see that also all the other properties are satisfied by our definition. Let $X_n$ be a finite subset of $X$. Given a distribution $P$ over $X_n$, let $\Pi = P \times P$ denote the distribution on the product space $X_n \times X_n$. For any embedding $f$, let $\widetilde{\rho_f}(u,v)$ denote the normalized ratio of distances given by

$$\widetilde{\rho_f}(u,v) := \frac{(n)(n-1)\rho_f(u,v)}{2\sum_{(u,v)\in\binom{X}{2}}\rho_f(u,v)}.$$

The $\sigma$-distortion is then defined as

$$\mathbb{E}_\Pi(\widetilde{\rho_f}(u,v) - 1)^2. \tag{5}$$

If $P$ is a uniform probability distribution over $X_n$, then $\sigma$-distortion measures the variance of the distribution of the normalized ratio of distances, $\widetilde{\rho_f}(u,v)$.

**Theorem 2 (Properties of $\sigma$-distortion).** *The $\sigma$- distortion (a) is **invariant to scale and translations**, (b) satisfies the property of **monotonicity**, (c) is **robust to outliers in data** and **outliers in distances**, and (d) **incorporates a probability distribution** into its evaluation.*

In addition to satisfying all of the aforementioned properties, the proofs of Abraham, Bartal, and Neiman, 2011 can be extended to show that one can embed any finite subset of a doubling metric space into constant dimensional Euclidean space (or any $l_p$ space) with $\mathcal{O}(1)$ distortion. So the $\sigma$-distortion also satisfies the nice-to-have property regarding constant dimensional embeddings with bounded distortion. The formal guarantees are given in the following two theorems:

**Theorem 3 (General metric spaces: embeddable with constant $\sigma$-distortion in $\log n$ dimensions).** *Given any finite sample $X_n = \{x_1, x_2, ..., x_n\}$ from an arbitrary metric space $(X, d_X)$ and a probability distribution $P$ on $X_n$, for any $1 \leq p < \infty$ there exists an embedding $f : X_n \to l_p^D$, where $D = \mathcal{O}(\log n)$ with $\sigma$-distortion $= \mathcal{O}(1)$.*

**Theorem 4 (Doubling metric spaces: embeddable with constant $\sigma$-distortion in constant dimensions).** *Given any finite sample $X_n = \{x_1, x_2, ..., x_n\}$ from a doubling metric space $(X, d_X)$ and a probability distribution $P$ on $X_n$, for any $1 \leq p < \infty$ there exists an embedding $f : X_n \to l_p^D$, where $D = \mathcal{O}(1)$ with $\sigma$-distortion $= \mathcal{O}(1)$.*

## 4 Experiments

We evaluate the behavior of various distortion measures by conducting experiments in two different settings: 1) Dimensionality reduction 2) A pipeline consisting of dimensionality reduction followed by classification. We start with simulated data for which we know all ground truth parameters. In order to generate datasets of dimension $D$, we sample each coordinate independently from a specified 1-dimensional distribution. Several different distributions such as Gaussian distribution, Gamma distribution, Beta distribution, Gaussian mixture distribution, Laakso Space (Bartal, Gottlieb, and Neiman, 2015) with many different parameter settings have been used to conduct the experiments. Embeddings are then generated by various dimensionality reduction algorithms. In particular, we

used Isomap (Tenenbaum, De Silva, and Langford, 2000), Maximum Variance Unfolding(Weinberger and Saul, 2006), Multidimensional Scaling, PCA (Hotelling, 1933), Probabilistic PCA (Tipping and Bishop, 1999), and Structure Preserving Embedding (Shaw and Jebara, 2009). All experiments have been repeated 10 times, the error bars in the plots depict the standard deviations over the different repetitions.

**Embedding dimension vs distortion:** For a dataset sampled from a Euclidean space with fixed dimension, it is natural to expect that any meaningful measure of distortion decreases with increasing embedding dimension. Intuitively, higher-dimensional spaces have more degrees of freedom to place points, and theoretical results confirm the general tendency (Chan, Gupta, and Talwar, 2010; Abraham, Bartal, and Neiman, 2008). In Figure 1, we show that this behavior can also be verified experimentally for most measures of distortion including $\sigma-$distortion, except for average distortion. The failure of average distortion to conform to this trend is clearly because it does not demonstrate invariance to scaling. To clarify, average distortion simply computes the sum of the ratios of distances $\rho_f(u, v)$. An embedding can deviate from isometry by either contracting the distances between pairs of points or expanding them relative to the original distances. An immediate consequence of scale invariance of a distortion is that it treats expansions and contractions symmetrically in the following sense: if an embedding expands all distances by $\alpha$, a scale invariant distortion would assign the embedding the same value of distortion as an embedding that contracts them by $\alpha$. Average distortion does not possess this property and places undue emphasis on expansions and underscores contractions. The balance of the weight of contractions and expansions influence the trend followed by average distortion, which is thereby unpredictable.

**Distortion vs original dimension:** For datasets generated from Euclidean spaces of increasing dimension, it is also natural to expect that for a fixed embedding dimension, the quality of an embedding decreases with increasing original dimension. The intuition here is that the data gets more complex, but the embedding space does not have the capacity to reflect this. To our surprise, this behavior cannot be observed for most of the traditional measures of distortion, as can be seen from Figures 2 and 3 (this observation was one of the starting points of this whole line of research). When looking closer into the data, we come to the conclusion that the reason for this failure is that these measures (except for average distortion, which shows erratic behaviour due to its dependence on the scale of the embedding) suffer from outliers, which disproportionately affect the distortion measures. There are only two measures of distortion that show the desired behavior: $\sigma$-distortion and $\epsilon$-distortion. We attribute this to the fact that these two measures are robust to outliers (as also been shown in our theoretical results). We can also see from the error bars in Figures 2, 3 that the variability of the rest of the distortion measures is significantly larger compared to that of $\epsilon$-distortion as well as $\sigma$-distortion. Again we attribute this behavior to the brittleness of the other distortions in the presence of outliers.

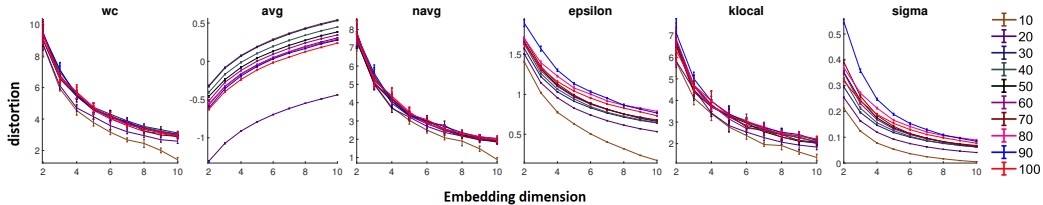

Figure 1: Embedding dimension vs various measures of distortion. From left to right: $\Phi_{wc}$, $\Phi_{avg}$, $\Phi_{navg}$, $\Phi_\epsilon$ for $\epsilon = 0.1$, $\Phi_{klocal}$ for k = 5, $\Phi_\sigma$. The color of each curve indicates the dimension of the original space, the x-axis the dimension of the embedding space. We can clearly see that for all but the average distortion, distortion decreases with embedding dimension. Data was generated according to a standard normal distribution of dimension as indicated by the color, embeddings have been generated using Isomap. Results for other distributions and algorithms look similar.

**Effect of noise:** In order to test the effect of noise on various measures of distortion, we generated mixture of Gaussian data in $\mathbb{R}^2$ similar to that of the previous experiment and added normally distributed noise in $\mathbb{R}^{20}$ of increasing variance to the data to generate different datasets. Embeddings were then performed using various algorithms into $\mathbb{R}^2$. In a first evaluation, we investigated whether the distortion increases with increasing noise. In Figure 4 (left) we can see that while $\sigma$-distortion clearly shows the desired trend, all other measures of distortion fail to show the correct behavior.

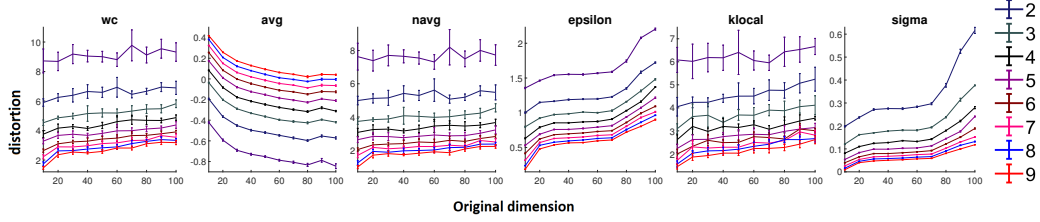

Figure 2: Original dimension vs measures of distortion. From left to right: $\Phi_{wc}$, $\Phi_{avg}$, $\Phi_{navg}$, $\Phi_\epsilon$ for $\epsilon = 0.1$, $\Phi_{klocal}$ for k = 5, $\Phi_\sigma$. The x-axis shows the dimension of the original space, the color of the curve corresponds to the dimension of the embedding space. Each curve corresponds to Isomap embeddings of data generated according to gamma distribution (a = 1.5, b = 4) from Euclidean space of dimensions $(10 : 10 : 100)$. Results for other distributions and algorithms look similar.

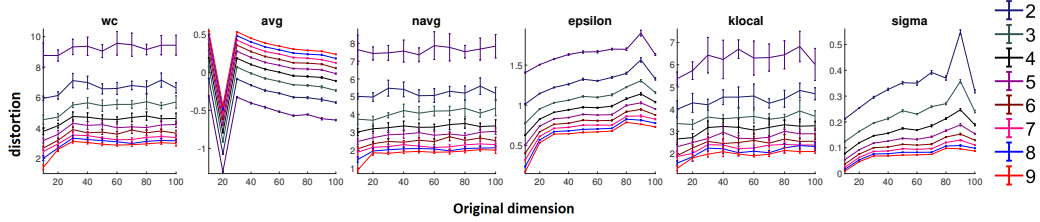

Figure 3: Same setting as in Figure 2, but data generated according to beta distribution (a = 0.75, b = 0.75).

In a second evaluation, we then performed classification on the embedded data. The corresponding SVM and kNN loss are plotted against the variance of the additive noise. Figure 4 clearly shows that the SVM and kNN classification loss increase with increasing variance of noise. This reiterates that the quality of the embedding indeed worsens with increasing additive noise. We performed this experiment using different embedding algorithms (Isomap, PCA, MVU) and the plots in all the experiments paint the same picture.

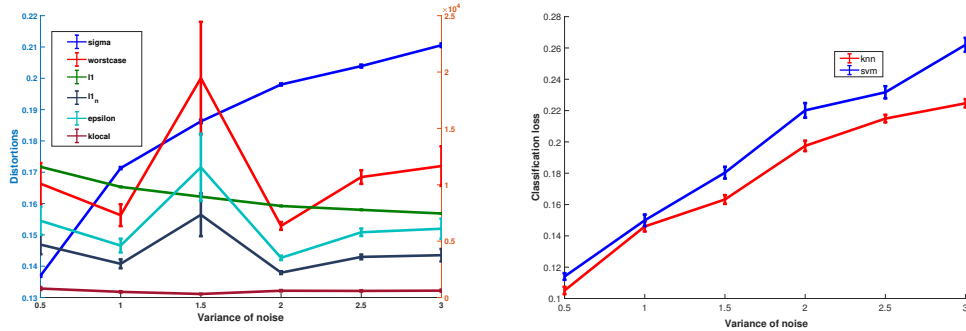

Figure 4: Left: Variance of noise vs distortion measures. The x-axis shows the variance of noise. As the measures of distortion are not all in the same range, we added two y-axes: the left (blue) one for $\sigma$-distortion, and the right (red) one for the values of the rest of the distortion measures. Right: Variance of noise vs. classification error. The x-axis shows the variance of noise, the y-axis the classification error. All embeddings here are created using Isomap. The behavior corresponding to the other embedding algorithms is similar.

**Distortion vs classification accuracy:** In this set of experiments, we want to investigate whether a measure of distortion is a good indicator for classification accuracy. To this end, we sampled data from various mixture of Gaussian distributions in $\mathbb{R}^2$ with different sets of parameters. Gaussian noise in $\mathbb{R}^{20}$ was then added to the data to generate various datasets. The datasets were then embedded into $\mathbb{R}^2$ using various embedding algorithms: PCA (Hotelling, 1933), GPLVM (Lawrence, 2004), Isomap (Tenenbaum, De Silva, and Langford, 2000), MVU (Weinberger and Saul, 2006), SPE (Shaw

and Jebara, 2009). Classification is performed on the resulting embeddings using kernel SVM (with RBF kernel) and kNN classification algorithms. In Figure 5, we plot the distortions incurred by the embeddings against the classification loss incurred by the classifier (where we sorted the outcome of all the simulations according to their resulting classification accuracy). Note that in this experiment, we compare the quality of embeddings across different embedding algorithms. The ideal behavior would be that distortion increases with increasing classification loss (in such a case, a measure of distortion could be used to select the best embedding, for example). This setting encapsulates the idea of using distortion measure as a means of evaluating the quality of an embedding in machine learning tasks. The plots clearly show that $\sigma$-distortion and $\epsilon$-distortion consistently show the expected increasing trend with the classification loss, whereas the other measures of distortion fail to do so.

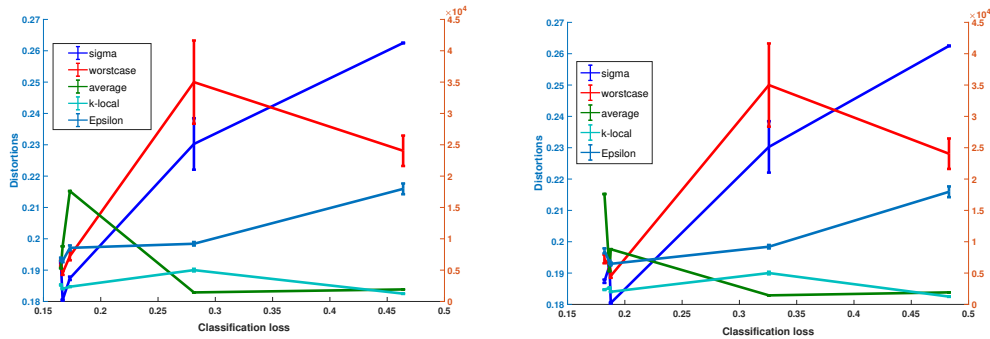

Figure 5: Classification error vs. distortion, for kNN (left) and SVM (right). x-axis: classification error and y-axis: distortion. The measures of distortion are not all in the same range, we added two y-axis: the left (blue) one for $\sigma$-distortion, and the right (red) one for the values of the rest of the distortion measures. Each curve corresponds to a distortion measure as indicated in the legend. The distortions are scaled appropriately for visualization.

## 5 Discussion

We investigate the properties of various measures of distortion for machine learning. Both in theory and experiments we can demonstrate that many of the existing measures of distortion behave in an undesired way: in simulations they show the wrong tradeoff with respect to the dimension of the original space, and they are not robust to noise or outliers, and cannot serve as a good indicator for classification accuracy. As an alternative, we define a new measure of distortion, called $\sigma$-distortion. In a nutshell, it measures the variance of the pairwise distortion ratios (rather than a norm of the vector of these ratios). We can show in theory and in experiments that it satisfies all our desirable properties. There is only one existing measure of distortion that comes close to our new $\sigma$-distortion, namely the $\epsilon$-distortion. It explicitly excludes an $\epsilon$ fraction of outlier points from the distortion computation. For most properties it behaves nice as well, but it fails to take the probability measure into account. This is important because $\epsilon$-distortion provides no guarantees on $\epsilon$ fraction of the pairwise distances, which could be critical for a given machine learning task. One of its drawbacks is that it has an important parameter to tune, the value of $\epsilon$ (fraction of outliers), whereas $\sigma$-distortion does not have a parameter. Our work clearly shows the need to study measures of distortion from a more systematic point of view, both in theory and practice.

### Acknowledgments

This work has been supported by the Institutional Strategy of the University of Tübingen (Deutsche Forschungsgemeinschaft, DFG, ZUK 63) and the International Max Planck Research School for Intelligent Systems (IMPRS-IS).

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
