[Supplementary Material]

# Supplement to "Measures of distortion for machine learning"

## A Proof of Theorems 1 and 2

**Homogeneous metric:** A metric, $d_X$, on a vector space, $X$, is said to satisfy the property of homogeneity if for all $u, v \in X$ and for any scalar $\alpha \in \mathbb{R}$, $d_X$ satisfies the following condition: $d_X(\alpha \cdot u, \alpha \cdot v) = |\alpha| \cdot d_X(u,v)$. We refer to such a metric, $d_X$, as a homogeneous metric and the corresponding metric space, $(X, d_X)$, as a homogeneous metric space.

**Translation invariant metric:** A metric $d_X$ on a vector space $X$ is said to be translation invariant if for any $u, v$ and $w \in X$, $d_X$ satisfies the following condition: $d_X(u + w, v + w) = d_X(u, v)$.

Recall that $(X, d_X)$ is an arbitrary metric space and $(Y, d_Y)$ is assumed to be a homogeneous and a translation invariant metric. Let $f, g : (X, d_X) \to (Y, d_Y)$ be two injective mappings. For any $S \subseteq \binom{X}{2}$, let $f_S$ denote the restriction of the mapping $f$ to the set $S$. Formally, $f_S := \{(f(u), f(v)) | (u, v) \in S\}$.

### A.1 Scale and translation invariance

A distortion measure $\Phi$ is said to be **scale invariant** if for any $\alpha \in \mathbb{R}$

$$\forall u \in X, f(u) = \alpha g(u); \implies \Phi(f) = \Phi(g).$$

$\Phi$ is said to be **translation invariant** if for any $y \in Y$,

$$\forall u \in X, f(u) = g(u) + y; \implies \Phi(f) = \Phi(g).$$

**(a) $\Phi_{wc}$, $\Phi_{navg}$, $\Phi_\epsilon$, $\Phi_{klocal}$ and $\Phi_\sigma$ are invariant to scaling:**

*Proof.* The proofs naturally follow from the definitions of these distortion measures and rely on the assumption of homogeneity of the target metric.

**worstcase distortion:** By the virtue of homogeneity of $d_Y$,

$$
\begin{aligned}
\Phi_{wc}(g) &= \max_{u \neq v \in X} \left\{ \frac{|\alpha| \cdot d_Y(f(u), f(v))}{d_X(u,v)} \right\} \cdot \max_{u \neq v \in X} \left\{ \frac{d_X(u,v)}{|\alpha| \cdot d_Y(f(u), f(v))} \right\} \\
&= \Phi_{wc}(f).
\end{aligned}
$$

**normalized average distortion:**

$$
\begin{aligned}
\Phi_{navg}(g) &= \frac{2}{n(n-1)} \sum_{u \neq v \in X} \frac{\rho_g(u,v)}{\min_{u \neq v \in X} \rho_g(u,v)}, \qquad \text{where } \rho_g(u,v) = \frac{d_Y(g(u), g(v))}{d_X(u,v)}. \\
\rho_g(u,v) &= \frac{|\alpha| \cdot d_Y(f(u), f(v))}{d_X(u,v)} = |\alpha| \cdot \rho_f(u,v). \quad \text{(from homogeneity of } d_Y\text{)}. \\
\Phi_{navg}(g) &= \frac{2}{n(n-1)} \sum_{u \neq v \in X} \frac{|\alpha| \cdot \rho_f(u,v)}{\min_{u \neq v \in X} |\alpha| \cdot \rho_f(u,v)} \\
&= \Phi_{navg}(f).
\end{aligned}
$$

**$\epsilon$-distortion:** The scale invariance of $\epsilon$-distortion follows from the fact that if $f, g$ are two embeddings that are scaled versions of each other, then for any subset $S$ of $\binom{X}{2}$, the worstcase distortions of $f_S, g_S$ are equal: For any $\epsilon \in (0, 1)$,

$$\Phi_\epsilon(g) = \min_{S \subset \binom{X}{2}, |S| \geq (1-\epsilon)\frac{n(n-1)}{2}} \Phi_{wc}(g_S)$$

and thus, for any $S \subset \binom{X}{2}$,

$$\Phi_{wc}(g_S) = \max_{\{\{u,v\} \in S\}} \left\{ \frac{d_Y(g(u), g(v))}{d_X(u,v)} \right\} \cdot \max_{\{\{u,v\} \in S\}} \left\{ \frac{d_X(u,v)}{d_Y(g(u), g(v))} \right\}$$

$$= \max_{\{\{u,v\} \in S\}} \left\{ \frac{|\alpha| \cdot d_Y(f(u), f(v))}{d_X(u,v)} \right\} \cdot \max_{\{\{u,v\} \in S\}} \left\{ \frac{d_X(u,v)}{|\alpha| \cdot d_Y(f(u), f(v))} \right\}$$

$$= \Phi_{wc}(f_S).$$

**k-local distortion:** For any $k \in \mathbb{N}$ and for any $u \in X$, let $kNN(u)$ denote the k-nearest neighbours of $u$ in $X$ according to $d_X$. The set $S$ defined as $\{\{u,v\} \mid u, v \in X, v \in kNN(u)\}$ is a subset of $\binom{X}{2}$. As shown in the case of $\epsilon$-distortion, for any two embeddings $f, g$ which are scaled versions of each other, we have $\Phi_{wc}(g_S) = \Phi_{wc}(f_S)$ and hence $\Phi_{klocal}(g) = \Phi_{klocal}(f)$.

**$\sigma$-distortion:** For any $\{u,v\} \in \binom{X}{2}$, let $\widetilde{\rho_f}(u,v)$ denote the normalized ratio of distances defined as $\rho_f(u,v)/\alpha_f$, where $\alpha_f = \sum\limits_{\{u,v\} \in \binom{X}{2}} \rho_f(u,v)/\binom{n}{2}$. Observe that, by the virtue of homogeneity of $d_Y$, $\widetilde{\rho_f}(u,v) = \widetilde{\rho_g}(u,v)$. Therefore,

$$\Phi_\sigma(g) = \mathbb{E}_\Pi(\widetilde{\rho_g} - 1)^2 = \mathbb{E}_\Pi(\widetilde{\rho_f} - 1)^2 = \Phi_\sigma(f).$$

$\square$

**(b) $\Phi_{avg}$ and $\Phi_{l_q}$ are not invariant to scaling.**

*Proof.* The proof follows from the linearity of the expectation. We prove the statement for the $l_q$ distortion, the case of the average distortion then is the special case of $q = 1$.

$$\Phi_{l_q}(g) = \mathbb{E}_\Pi(\rho_g(u,v)^q)^{\frac{1}{q}} = \mathbb{E}_\Pi(|\alpha| \cdot \rho_f(u,v)^q)^{\frac{1}{q}} = |\alpha| \cdot \Phi_{l_q}(f).$$

$\square$

**(c) All the distortion measures, $\Phi_{wc}, \Phi_{l_q}, \Phi_{navg}, \Phi_\epsilon, \Phi_{klocal}$ and $\Phi_\sigma$ are invariant to translations.**

*Proof.* It is straightforward to see that all the distortion measures derive this property from the translation invariance of $d_Y$. $\square$

## A.2 Monotonicity

For any $f : (X, d_X) \to (Y, d_Y)$, let $\alpha_f = \binom{n}{2} / \sum\limits_{u \neq v \in X} \rho_f(u,v)$. Observe that $\alpha_f > 0$. Let $f, g : (X, d_X) \to (Y, d_Y)$ be two embeddings such that for all $u, v \in X$,

$$\alpha_f \cdot \rho_f(u,v) \leq \alpha_g \cdot \rho_g(u,v) \leq 1 \quad \text{or} \quad \alpha_f \cdot \rho_f(u,v) \geq \alpha_g \cdot \rho_g(u,v) \geq 1.$$

**(a) $\Phi_{wc}, \Phi_{navg}, \Phi_\epsilon, \Phi_{klocal}$ and $\Phi_\sigma$ satisfy the property of monotonicity.**

*Proof.* The proofs follow directly from the definitions and utilize the scale invariance property of the corresponding distortion measure:

**worstcase distortion:**

$$\Phi_{wc}(f) = \Phi_{wc}(\alpha_f \cdot f) \qquad\qquad \text{(Scale invariance of } \Phi_{wc})$$

$$= \max_{u \neq v \in X} \{\alpha_f \cdot \rho_f(u,v)\} \cdot \frac{1}{\min\limits_{u \neq v \in X} \{\alpha_f \cdot \rho_f(u,v)\}}$$

$$\geq \max_{u \neq v \in X} \{\alpha_g \cdot \rho_g(u,v)\} \cdot \frac{1}{\min\limits_{u \neq v \in X} \{\alpha_g \cdot \rho_g(u,v)\}}$$

$$= \Phi_{wc}(g).$$

The inequality follows since $\max\limits_{u\neq v\in X}\{\alpha_f\cdot\rho_f(u,v)\}\geq 1$ and $\min\limits_{u\neq v\in X}\{\alpha_f\cdot\rho_f(u,v)\}\leq 1$.

**normalized average distortion:**

$$\Phi_{navg}(f)=\Phi_{navg}(\alpha_f\cdot f)\qquad\qquad\qquad\text{(Scale invariance of }\Phi_{navg}\text{)}$$

$$=\frac{\sum\limits_{u\neq v\in X}\{\alpha_f\cdot\rho_f(u,v)\}}{\min\limits_{u\neq v\in X}\{\alpha_f\cdot\rho_f(u,v)\}}$$

$$\geq\frac{2}{(n)\cdot(n-1)\min\limits_{u\neq v\in X}\{\alpha_g\cdot\rho_g(u,v)\}}$$

$$=\Phi_{navg}(g).$$

The inequality follows since $\min\limits_{u\neq v\in X}\{\alpha_f\cdot\rho_f(u,v)\}\leq 1$.

**$\epsilon$-distortion:** Let $\Psi=\mathop{\arg\min}\limits_{S\subset\binom{X}{2},|S|>(1-\epsilon)\frac{n(n-1)}{2}}\Phi_{wc}(f_S)$. Then,

$$\Phi_\epsilon(f)=\Phi_{wc}(f_\Psi)\geq\Phi_{wc}(g_\Psi)\geq\min\limits_{S\subset\binom{X}{2},|S|>(1-\epsilon)\frac{n(n-1)}{2}}\Phi_{wc}(g_S)=\Phi_\epsilon(g).$$

The first inequality follows due to the monotonicity of $\Phi_{wc}$.

**k-local distortion:** For any $k\in\mathbb{N}$, the set $S=\{(u,v)|u,v\in X,v\in kNN(u)\}$ is a subset of $\binom{X}{2}$. Then,

$$\Phi_{klocal}(f)=\Phi_{wc}(f_S)\geq\Phi_{wc}(g_S)=\Phi_{klocal}(g).$$

The first equality follows by definition and the first inequality follows due to the monotonicity of $\Phi_{wc}$.

**$\sigma$-distortion** For any $f:(X,d_X)\to(Y,d_Y)$ and for all $u\in X$, let $f'(u)=\frac{(n)(n-1)f(u)}{2\sum\rho_f(u,v)}$. Then

$$\Phi_\sigma(f)=\mathbb{E}_\Pi(\rho_{f'}(u,v)-1)^2.$$

Observe that, by the definition of monotonicity, we have that for all $u,v\in X$, $(\rho_{f'}(u,v)-1)^2\geq(\rho_{g'}(u,v)-1)^2$. Hence it follows that,

$$\Phi_\sigma(f)=\mathbb{E}_\Pi(\rho_{f'}(u,v)-1)^2)\geq\mathbb{E}_\Pi(\rho_{g'}(u,v)-1)^2)=\Phi_\sigma(g).$$

$\square$

**(b) $\Phi_{avg},\Phi_{l_q}$ fail to satisfy the property of monotonicity.**

*Proof.* Proof by contradiction.

**average distortion, $l_q$ distortion:**

Let $f,g:(X,d_X)\to(Y,d_y)$ be two embeddings such that there exists a constant $\beta>1$ and for all $u\in X$, $\beta f(u)=g(u)$. Let $\alpha_f=(n)(n-1)/2\sum\limits_{(u\neq v\in X)}\rho_f(u,v)$. Then for all $\{u,v\}\in\binom{X}{2}$, $f,g$ satisfy the following condition:

$$\alpha_f\rho_f(u,v)\leq\alpha_g\rho_g(u,v)\leq 1\quad\text{or}\quad\alpha_f\rho_f(u,v)\geq\alpha_g\rho_g(u,v)\geq 1.$$

However $\Phi_{avg}(f)=\frac{\Phi_{avg}(f)}{\beta}<\Phi_{avg}(g)$.
$\square$

## A.3 Robustness to outliers

### (a) $\Phi_{avg},\Phi_\epsilon$ and $\Phi_\sigma$ are robust to outliers in distances.

*Proof.* Let $X,d_X$ be an arbitrary metric space. Let $X_D=\{x_1,x_2,....,\}$ be a subset of $X$. Let $f:X_D\to X$ be an injective mapping such that there exists a constant $K\in\mathbb{N}$ such that the set $G=\left\{\{u,v\}\in\binom{X_D}{2}\mid d_X(f(u),f(v))\neq d_X(u,v)\right\}$ satisfies $|G|\leq K$. Let $f_n$ denote the

restriction of $f$ to $X_n = \{x_1, x_2, ..., x_n\}$. Let $I : (X, d_X) \to (X, d_X)$ be an isometry and let $I_n$ denote the restriction of $I$ to $X_n$.

**average distortion:** For any $n \in \mathbb{N}$, let $G_n = \left\{\{u, v\} \in \binom{X_n}{2} : d_X(f_n(u), f_n(v)) \neq d_X(u, v)\right\}$. By definition, it follows that for all $n \in \mathbb{N}$, there exists a $K \in \mathbb{N}$ such that, $|G_n| \leq |G| \leq K$. This implies that there exists a $n_0 \in \mathbb{N}$ such that, for all $n \geq n_0$, $G_n = G$ and $|G| = K$.

Therefore for any fixed $n \geq n_0$, $\rho_{f_n}$ can be expressed as

$$\rho_{f_n} = [\underbrace{1, 1, ..., 1}_{\binom{n}{2} - K \text{ times}}, \underbrace{\alpha_1, \alpha_2, ..., \alpha_K}_{K}],$$

where for all $i = \{1, 2, ..., K\}$, $\alpha_i$'s denote the ratio of distances $\rho_f(u, v)$ corresponding to each $\{u, v\} \in G$. Therefore for any fixed $n \geq n_0$, average distortion of $f_n$ can be expressed as

$$\Phi_{avg}(f_n) = \frac{(\binom{n}{2} - K) + \sum_{i=1}^{K} \alpha_i}{\binom{n}{2}}.$$

Hence $\lim_{n \to \infty} \Phi_{avg}(f_n) = 1 = \lim_{n \to \infty} \Phi_{avg}(I_n)$.

**$\epsilon$-distortion:** For any $\epsilon \in (0, 1)$,

$$\Phi_\epsilon(f_n) = \min_{S \subset \binom{X_n}{2}, |S| \geq (1-\epsilon)\frac{n(n-1)}{2}} \Phi_{wc}(f_{n_S}).$$

Observe that for all $n > 1 + \sqrt{1 + \frac{8K}{\epsilon}}$, there exists a set $S \subset \binom{X_n}{2}$ such that $|S| \geq (1 - \epsilon)\frac{n(n-1)}{2}$ and $\Phi_{wc}(f_{n_S}) = 1$. Hence $\lim_{n \to \infty} \Phi_\epsilon(f_n) = 1 = \lim_{n \to \infty} \Phi_\epsilon(I_n)$.

**$\sigma$-distortion:** The proof follows along the same lines as that of average distortion.

For any $f_n : (X_n, d_X) \to (X, d_X)$, $\Phi_\sigma(f_n)$ can be expressed as

$$\Phi_\sigma(f_n) = \frac{\sum_{(u,v) \in \binom{X_n}{2}} [\binom{n}{2}\rho_{f_n}(u, v) - \sum_{(u,v) \in \binom{X_n}{2}} \rho_{f_n}(u, v)]^2}{\binom{n}{2}[\sum_{(u,v) \in \binom{X_n}{2}} \rho_{f_n}(u, v)]^2}. \tag{1}$$

For any $n \in \mathbb{N}$, let $G_n = \{(u, v) \in X_n : d_X(f_n(u), f_n(v)) \neq d_X(u, v)\}$. By definition, it follows that for all $n \in \mathbb{N}$, there exists a $K \in \mathbb{N}$ such that, $|G_n| \leq |G| \leq K$. This implies that there exists a $n_0 \in \mathbb{N}$ such that, for all $n \geq n_0$, $G_n = G$ and $|G| = K$. Therefore for any fixed $n \geq n_0$, $\rho_{f_n}$ can be expressed as

$$\rho_{f_n} = [\underbrace{1, 1, ..., 1}_{\binom{n}{2} - K \text{ times}}, \underbrace{\alpha_1, \alpha_2, ..., \alpha_K}_{K}],$$

where for all $i = \{1, 2, ..., K\}$, $\alpha_i$'s denote the ratio of distances $\rho_f(u, v)$ corresponding to each $\{u, v\} \in G$. Therefore for any fixed $n \geq n_0$, $\sigma$-distortion of $f_n$ can be expressed as

$$\Phi_\sigma = \frac{\binom{n}{2}(K - \sum_{i=1}^{K} \alpha_i)^2(\binom{n}{2} - K) + \sum_{i=1}^{K}(\binom{n}{2}(\alpha_i - 1) + K - \sum_{i=1}^{K} \alpha_i)^2}{\binom{n}{2}(\binom{n}{2} - K + \sum_{i=1}^{K} \alpha_i)^2}.$$

Hence, $\lim_{n \to \infty} \Phi_\sigma(f_n) = 0 = \lim_{n \to \infty} \Phi_\sigma(I_n)$. $\square$

**(b) $\Phi_{avg}, \Phi_\epsilon$ and $\Phi_\sigma$ are robust to outliers in data**

*Proof.* The proofs follow by definition and utilize the subadditivity of the target metric $d_Y$.

**average distortion:** Let $f_n : X_n \to X$ be an embedding as specified in the premise of the definition. Then the average distortion of $f_n$ is evaluated as

$$\Phi_{avg} = \frac{\binom{n}{2} - (n-1) + \sum\limits_{i=1}^{n-1} \alpha_i}{\binom{n}{2}} \text{, where } \alpha_i = \frac{d_Y(f(x_i), f(x_0))}{d_X(x_i, x_0)} = \frac{d_Y(f(x_i), f(x_0))}{d_Y(f(x_i), I(x_0))}.$$

From the subadditivity of $d_Y$, it follows that

$$\left|\frac{d_Y(I(x_0), f(x_0))}{d_Y(I(x_0), f(x_i))} - 1\right| < \alpha_i < \frac{d_Y(I(x_0), f(x_0))}{d_Y(I(x_0), f(x_i))} + 1.$$

By construction, we have that $d_Y(I(x_0), f(x_i)) = d_Y(I(x_0), I(x_i)) = d_X(x_0, x_i) > \beta$ for some $\beta > 0$ and it follows that

$$0 < \left|\frac{d_Y(I(x_0), f(x_0))}{d_Y(I(x_0), f(x_i))} - 1\right| < \alpha_i < \frac{d_Y(I(x_0), f(x_0))}{\beta} + 1.$$

This implies,

$$\frac{\binom{n}{2} - (n-1)}{\binom{n}{2}} < \Phi_{avg}(f_n) < \frac{\binom{n}{2} - (n-1) + \frac{(n-1)d_Y(I(x_0), f(x_0))}{\beta}}{\binom{n}{2}}.$$

Note that $\beta > 0$ is a constant and for a fixed $x_0 \in X$, $d_Y(I(x_0), f(x_0))$ is also a constant. Therefore, $\lim\limits_{n \to \infty} \Phi_{avg}(f_n) = 1 = \lim\limits_{n \to \infty} \Phi_{avg}(I_n)$.

### $\epsilon$-distortion:

Let $f_n : X_n \to X$ be an embedding as specified in the premise of the definition. Its easy to verify that for any $\epsilon \in (0, 1)$ and for all $n > \frac{8}{\epsilon}$, there exists a set $S \subset \binom{X}{2}$ such that, $|S| > (1 - \epsilon)\frac{n(n-1)}{2}$ and the worstcase distortion of $f_n$ restricted to $S$ is $1 = \Phi_\epsilon(I_n)$.

### $\sigma$-distortion:

Let $f_n : X_n \to X$ be an embedding as specified in the premise of the definition. Let $\mu(\rho)$ denote $\sum\limits_{u \neq v \in X} \rho_{f_n}(u, v) / \binom{n}{2}$ and for all $i \in [n]$, let $\alpha_i = d_Y(f(x_i), f(x_0))/d_X(x_i, x_0)$. Then $\Phi_\sigma(f_n)$ is evaluated as

$$\frac{\sum\limits_{(u,v) \in \binom{X_n}{2}} [\rho_{f_n}(u, v) - \mu(\rho)]^2}{\binom{n}{2}\mu(\rho)^2} = \frac{\sum\limits_{i=1}^{\binom{n}{2}-(n-1)} [1 - \mu(\rho)]^2 + \sum\limits_{i=1}^{n-1} [\alpha_i - \mu(\rho)]^2}{\binom{n}{2}[\mu(\rho)]^2}.$$

By substituting $\mu(\rho) = \frac{\binom{n}{2}-(n-1)+\sum\limits_{i=1}^{n-1}\alpha_i}{\binom{n}{2}}$ we have

$$\Phi_\sigma(f_n) = \frac{\binom{n}{2}^2 (\sum\limits_{i=1}^{n-1} \alpha_i^2) - (2\binom{n}{2}(\binom{n}{2} - (n-1)))\sum\limits_{i=1}^{n-1} \alpha_i - \binom{n}{2}(\sum\limits_{i=1}^{n-1} \alpha_i)^2}{\binom{n}{2}[(\binom{n}{2} - (n-1))^2 + (\sum\limits_{i=1}^{n-1} \alpha_i)^2 + 2\sum\limits_{i=1}^{n-1} \alpha_i(\binom{n}{2} - (n-1))]^2}.$$

Since $I$ is an isometry, and since we have $f(x_i) = I(x_i)$ by definition, we obtain that

$$\frac{d_Y(f(x_i), f(x_0))}{d_X(x_i, x_0)} = \frac{d_Y(f(x_i), f(x_0))}{d_Y(I(x_i), I(x_0))} = \frac{d_Y(f(x_i), f(x_0))}{d_Y(f(x_i), I(x_0))}.$$

By subadditivity of $d_Y$, we have that, $\forall i \in [n]$

$$\left|\frac{d_Y(I(x_0), f(x_0))}{d_Y(I(x_0), f(x_i))} - 1\right| < \alpha_i < \frac{d_Y(I(x_0), f(x_0))}{d_Y(I(x_0), f(x_i))} + 1.$$

By construction, we have that $d_Y(I(x_0), f(x_i)) = d_Y(I(x_0), I(x_i)) = d_X(x_0, x_i) > \beta$ for some $\beta > 0$ and it follows that

$$0 < |\frac{d_Y(I(x_0), f(x_0))}{d_Y(I(x_0), f(x_i))} - 1| < \alpha_i < \frac{d_Y(I(x_0), f(x_0))}{\beta} + 1.$$

Hence there exists a constant $C \in \mathbb{R}$ such that for all $i \in \mathbb{N}$, $0 < \alpha_i < C$. It follows that

$$0 < \sum_{i=1}^{n-1} \alpha_i < (n-1)C, \ 0 < \sum_{i=1}^{n-1} \alpha_i^2 < (n-1)C^2 \text{ and } 0 < (\sum_{i=1}^{n-1} \alpha_i)^2 < ((n-1)C)^2.$$

By substitution and simplification, it follows that $\forall n > 2$,

$$\Phi_\sigma(f_n) < \frac{\binom{n}{2}^2 C^2 (n-1)}{\binom{n}{2}(\binom{n}{2} - (n-1))^2} \text{ and}$$

$$\Phi_\sigma(f_n) > \frac{-(2\binom{n}{2}(\binom{n}{2} - (n-1)))((n-1)C) - \binom{n}{2}((n-1)C)^2}{\binom{n}{2}[(\binom{n}{2} - (n-1))^2 + ((n-1)C)^2 + 2(n-1)C(\binom{n}{2} - (n-1))]^2}.$$

Observe that

$$\lim_{n\to\infty} \frac{-(2\binom{n}{2}(\binom{n}{2} - (n-1)))((n-1)C) - \binom{n}{2}((n-1)C)^2}{\binom{n}{2}[(\binom{n}{2} - (n-1))^2 + ((n-1)C)^2 + 2(n-1)C(\binom{n}{2} - (n-1))]^2} = 0 \text{ and}$$

$$\lim_{n\to\infty} \frac{\binom{n}{2}^2 C^2 (n-1)}{\binom{n}{2}(\binom{n}{2} - (n-1))^2} = 0.$$

Hence it follows that $\lim_{n\to\infty} \Phi_\sigma(f_n) = 0 = \lim_{n\to\infty} \Phi_\sigma(I_n)$. $\qquad\square$

**(c) $\Phi_{wc}$, $\Phi_{navg}$ and $\Phi_{klocal}$ are not robust to outliers in data or distances.**

*Proof.* Proof by a counterexample. We first construct a sequence of embeddings for which the number of distances that are distorted is bounded by a constant $K$ and the distances distorted stem from mapping a single point away from an isometry. Then we show that in the limit, as the size of the metric space tends to infinity, $\Phi_{wc}$, $\Phi_{navg}$ and $\Phi_{klocal}$ do not have the same distortion as that of an isometry.

Let $\{e_i\}_{\{i=1,..,d\}}$ denote the standard orthonormal basis for $(\mathbb{R}^d, l_2)$. Fix $x_0 = (\alpha - 1) \cdot e_d$ for some $1 < \alpha < 2$. Let $x_1 = e_d$. For any $n \in \mathbb{N}$, set $X_n = \{x_2, x_3, ..., x_n\}$, such that for all $i \in \{2, 3, ..., n\}$, $x_i$ is sampled according to some distribution $\mathcal{P}$ on $span\{e_1, ..., e_{d-1}\}$. Let $f : (\mathbb{R}^d, l_2) \to (\mathbb{R}^d, l_2)$ be the mapping defined as $f(x) = x$, $\forall x \in \mathbb{R}^d$, $x \neq x_0$ and $f(x_0) = -x_0$. Let $f_n$ denote the mapping $f$ restricted to $X_n \cup \{x_0, x_1\}$. It is easy to verify that the ratio of distances $\rho_{f_n}(x_0, x_1) = \alpha$ and $\rho_{f_n}(x_i, x_j) = 1$ for any $\{x_i, x_j\} \in \binom{X_n}{2} \setminus \{x_0, x_1\}$.

**worstcase distortion:**

The worstcase distortion evaluated on $f_n : (X_n \cup \{x_0, x_1\}, d_X) \to (X, d_X)$ for any $n \in \mathbb{N}$ is $\alpha$ and thus $\lim_{n\to\infty} \Phi_{wc}(f_n) = \alpha > \lim_{n\to\infty} \Phi_{wc}(I_n) = 1$, where $I_n$ denotes the restriction of the mapping $I$ to $X_n \cup \{x_0, x_1\}$.

**normalized average distortion:**

The sequence of average distortions evaluated on mappings $\{f_n\}$ is given by:

$$\Phi_{navg}(f_n) = \frac{(\binom{n}{2} - 1)\alpha + 1}{\binom{n}{2}}$$

Thus, $\lim_{n\to\infty} \Phi_{navg}(f_n) = \alpha > \Phi_{navg}(I_n) = 1$.

**k-local distortion:**

Since $x_0$ lies in the set of k-nearest neighbours of $x_1$, for any $n \geq 2$, the k-local distortion evaluated on the mapping $f_n : (X_n \cup \{x_0, x_1\}, d_X) \to (X, d_X)$ is $\alpha$ and thus $\lim_{n\to\infty} \Phi_{klocal}(f_n) = \alpha > \lim_{n\to\infty} \Phi_{klocal}(I_n) = 1$. $\qquad\square$

## A.4 Incorporation of a probability distribution

Let $(X, d_X)$ be an arbitrary metric space. Let $X_n = \{x_1, x_2, ..., x_n\}$ be a finite subset of $X$. Let $P$ denote a probability distribution on $X_n$ and let $\Pi = P \times P$ denote the product distribution on $X_n \times X_n$. Fix any arbitrary $x^*, y^* \in X_n$ such that $P(x^*) > P(y^*)$. Let $x', y' \in X$ such that for all $i \in [n]$, $d_X(x_i, x') = \alpha d_X(x_i, x^*)$ and $d_X(x_i, y') = \alpha d_X(x_i, y^*)$. Let $f, g : X_n \to X$ be two embeddings such that:

$$f(x) = \begin{cases} x', & \text{if } x = x^*. \\ x, & \text{otherwise.} \end{cases} \quad , \quad g(x) = \begin{cases} y', & \text{if } x = y^*. \\ x, & \text{otherwise.} \end{cases}$$

**(a) $\Phi_{wc}, \Phi_{avg}, \Phi_{navg}, \Phi_\epsilon$, and $\Phi_{klocal}$ fail to incorporate a probability distribution into their evaluation.**

*Proof.* The proofs follow directly from the definitions of the distortion measures.

**worstcase, normalized avg, $\epsilon$-distortion and k-local distortion:** The above distortion measures, by definition are independent of the probability distribution over the data space.

**average $(l_q)$ distortion:** The proof follows from explicit evaluation of the average distortion of any two embeddings $(f, g)$ that satisfy the conditions as specified in the definition. Average distortion of $f$ can be expressed as:

$$\Phi_{avg}(f) = \mathbb{E}_\Pi[\rho_f(u, v)] = [\sum_{i,j \neq 1,2} \Pi_{ij} + \sum_{i \neq 1} \alpha \Pi_{i1} + \sum_{i \neq 2} \Pi_{i2}].$$

$$\Phi_{avg}(f) - \Phi_{avg}(g) = [\sum_{i \neq 1,2} (\Pi_{i1} - \Pi_{i2})(\alpha - 1)]$$

$$= (\sum_{i \neq 1,2} (\Pi_i)(\Pi_1 - \Pi_2)(\alpha - 1))$$

$$< 0 \quad \text{if } \alpha < 1.$$

The case of $l_q$-distortion follows similarly. $\square$

**(b) $\Phi_\sigma$ incorporates a probability distribution into its evaluation.**

*Proof.* Observe that $\sum_{u \neq v \in X} \rho_f(u, v) = \sum_{u \neq v \in X} \rho_g(u, v) = \binom{n}{2} - (n - 1) + (n - 1)\alpha$. Let $\kappa = \binom{n}{2} / \sum_{u \neq v \in X} \rho_f(u, v) = \binom{n}{2} / \sum_{u \neq v \in X} \rho_g(u, v)$. Then,

$$\Phi_\sigma(f) - \Phi_\sigma(g) = \mathbb{E}_\Pi[(\kappa \rho_f(u, v) - 1)^2] - \mathbb{E}_\Pi[(\kappa \rho_g(u, v) - 1)^2]$$

$$= \mathbb{E}_\Pi[\kappa^2 \rho_f(u, v)^2 - 2\kappa \rho_f(u, v)] - \mathbb{E}_\Pi[\kappa^2 \rho_g(u, v)^2 - 2\kappa \rho_g(u, v)].$$

$$= \kappa^2(\alpha^2 - 1) \sum_{i,j \neq 1,2} (\Pi_{1i} - \Pi_{2i}) - 2\kappa(\alpha - 1) \sum_{i,j \neq 1,2} (\Pi_{1i} - \Pi_{2i})$$

$$= K^2(\Pi_1 - \Pi_2)(\alpha - 1)(\kappa(\alpha + 1) - 2) \sum_{i \neq 1,2} \Pi_i.$$

It is easy to verify that for any $\alpha \geq 0$ and for all $n > 4$, $(\kappa(\alpha + 1) - 2)(\alpha - 1) \geq 0$ and by definition $\Pi_1 > \Pi_2$. Therefore, $\Phi_\sigma(f) \geq \Phi_\sigma(g)$. $\square$

# B Proofs of Theorems 3 and 4

**Doubling space:** A metric space $(X, d_X)$ is referred to as a doubling space if there exists a doubling constant $\lambda > 0$ such that for any $u \in X$ and $r > 0$, the ball $B(u, r) = \{v \mid d_X(u, v) < r\}$ can be covered by at most $\lambda$ balls of radius $r/2$.

## B.1 Theorem 3:

Abraham, Bartal, and Neiman, 2011 showed the existence of an embedding from any arbitrary metric space into an Euclidean space with properties as stated in Theorem A. Combined with Lemma A, this provides the sought upper bound on $\sigma$-distortion.

**Theorem A.** *(Abraham, Bartal, and Neiman, 2011) Given any arbitrary finite metric space $(X, d_X)$, there exists an embedding $f : X \to l_p^D$ where $D = O(\log n)$ and for any $\epsilon \in (0, 1)$, there exists a set $G_\epsilon$ and constants $C_1$ and $C_2$ (independent of $\epsilon$,n) such that $|G_\epsilon| \geq (1 - \epsilon) \cdot \binom{|X|}{2}$ and for any $x, y \in G_\epsilon$: $C_1 \leq \frac{\|f(x)-f(y)\|_p}{d_X(x,y)} \leq C_2 \cdot \log(\frac{2}{\epsilon})$.*

**Lemma A.** *(Abraham, Bartal, and Neiman, 2011) Given any finite metric spaces $(X, d_X)$ and $(Y, d_Y)$ and an embedding $f : X \to Y$ satisfying properties described in Theorem A, for any distribution $\Pi$ over $X \times X$, there exists a constant $K = K(\Pi)$ such that $\mathbb{E}_\Pi \left( \frac{d_Y(f(x),f(y))}{d_X(x,y)} \right)^2 < K$ and $\mathbb{E}_\Pi \left( \frac{d_Y(f(x),f(y))}{d_X(x,y)} \right) < K$.*

*Proof of Theorem 3:* From theorem A, by choosing any $\epsilon < \frac{2}{n(n-1)}$, we have that $\forall x, y \in X$, $C_1 \leq \frac{\|f(x)-f(y)\|_p}{d_X(x,y)}$ implies $C_1 \leq \mathbb{E}_\Pi \left( \frac{\|f(x)-f(y)\|_p}{d_X(x,y)} \right)$. Recall that $\sigma$-distortion is defined as

$$\frac{\mathbb{E}_\Pi[\rho_f(u,v) - \frac{2}{n(n-1)}\sum \rho_f(u,v)]^2}{(\frac{2}{n(n-1)}\sum \rho_f(u,v))^2}.$$

Combined with Lemma A, this completes the proof of Theorem 3. $\square$

## B.2 Theorem 4:

(Abraham, Bartal, and Neiman, 2011) also showed the existence of an embedding from any arbitrary doubling metric space into an Euclidean space with properties as stated in Theorem B and Lemma B. These results provide an upper bound on the $\sigma$-distortion evaluated on this embedding.

**Theorem B.** *(Abraham, Bartal, and Neiman, 2011) Given any finite metric space $(X, d_X)$ with doubling constant $\lambda$, there exists an embedding $f : X \to l_p^D$ and a constant $K = K(\lambda)$ such $D < K$ and for any $\epsilon \in (0, 1)$, there exists a set $G_\epsilon$ and constants $C_1$ and $C_2$ (independent of $\epsilon$) such that $|G_\epsilon| \geq (1 - \epsilon) \cdot \binom{|X|}{2}$ and for any $x, y \in G_\epsilon$: $C_1 \leq \frac{\|f(x)-f(y)\|_p}{d_X(x,y)} \leq C_2 \cdot \log^{26}(\frac{1}{\epsilon})$.*

**Lemma B.** *(Abraham, Bartal, and Neiman, 2011) Given a finite metric space $(X, d_X)$ with doubling constant $\lambda$, another metric space $(Y, d_Y)$ and an embedding $f : X \to Y$ satisfying the properties described in Theorem B, then for any distribution $\Pi$ over $X \times X$, there exists a constant $K = K(\Pi)$ such that $\mathbb{E}_\Pi \left( \frac{d_Y(f(x),f(y))}{d_X(x,y)} \right)^2 < K$.*

*Proof of Theorem 4:* Same as the proof of Theorem 4. $\square$