[Reviews · NeurIPS 2018]

Reviewer 1



The paper states a list of desirable properties for distortion measure for embeddings including scale and shift invariance, robust handling of outliers and noise. They then proposes a specific measure that satisfies those properties: the measure essentially looks at the variance of the standard distortion over randomly chosen pair of points. They show that this measure satisfies all the properties. They show that for any metric space can be embedded into O(log n) dimensional euclidean space with constant distortion under their new definition. For a metric space with constant doubling dimension they show that any finite set of sampled points can be embedded with constant distortion into constant dimensional Eucliean space.They also show using experiments that their distortion measure is better than other standard ones. The writing and the presentation is fair, but the novelty and strength of the contribution seems low for this conference. I was unable to open the supplementary material. It looks like the experiments are only done over synthetically generated data. To demonstrate the utility of the measure it would have been better to show this in some way over real data sets.

Reviewer 2



The paper discusses the downstream inference effects of embedding. The paper discusses several properties that are desirable, and provide status of many different embeddings. The paper also shows a thorough list of experiments to make its point. Overall the paper is a welcome addition to a recipe/faith/herd driven climate where "embedding X is the new thing and will solve this problem (never mind we forgot to find out if the embedding made sense at all". From a foundation perspective, the limit statements about embeddings were interesting. One comment in this context is that most embeddings (that work) are probabilistic. This is not the same as the discussion in lines 123 and the sequel, we are discussing choosing a random map from a collection of maps. So what would be limit analogue in the probability context? Describing that would be a valuable contribution both in concepts and experiments. One other comment is regarding outliers. This part is less clearly defined because the definition of outlier is not clear --- the authors themselves were compelled to make the distinction with outliers in distance etc.,

Reviewer 3



=== SUMMARY === When points in one metric space are embedded into another (eg. from high to lower dimension, or from a "difficult" to "easy" metric), the quality of the embedding is typically measured by some notion of distance distortion. This paper is a systematic study of distortion measures. It formally defines several desired properties of a distortion measure, and compares existing distortion measures from the lens of those formal definitions and by simulations on synthetic data. Based on these, the pros and cons of each measure are discussed, and a new notion of distortion is suggested. === COMMENTS === The systematic study of desirable properties of distortion is solid and in my view constitutes the main strength of this submission. The discussion leading to the formal definitions is clear and well-motivated, and the definitions are well-thought and display theoretical expertise. The findings, both in theory and in simulation, are interesting and for me they shed light on how to interpret embedding results. The new sigma-distortion is simple and straightforward, but it is interesting to see both rigorously and experimentally how it compares favorably to existing distortion measures (which are also simple and straightforward, but apparently behave less desirably). Since the rest of my review will be mostly critical, I will say in advance that the strengths mentioned so far are sufficient for me to view this paper as a solid contribution that can be accepted. The main weakness of the paper is a somewhat tenuous connection to machine learning in terms of motivation, application and literature. Motivation: While the paper claims to be about ML, the relevance to ML is described only superficially and in broad strokes. It is true that the standard ML pipeline embeds the data into Euclidean space, but mostly the domain space is not metric, but a space of "real-world objects" (images, text documents etc) with no a-priori well-defined or quantifiable measure of distance, and therefore the embedding cannot be evaluated in terms of distortion (this "embedding" is actually feature generation). Distortion measures are relevant specifically for dimensionality reduction, either to achieve computational tractability or when the data is assumed to have low intrinsic dimension despite its high ambient dimension, which is quite a common assumption and well-observed phenomenon. Another possibly relevant case is when the data is endowed with a distance function, but it is "difficult" and one wants to embed it into a nicer space like Euclidean which is easier to work with. I am not sure I am giving a good overview of the ML motivation; my point is that being more specific about the relevance of the subject (preferably with appropriate citations) would help the paper. Literature: The introduction and the list of allegedly "popular" distortion measures cite prior theoretical work, but not any ML work at all. (Are there any ML citations before the experimental section?) It seems that the authors are well-versed in the mathematical literature, but could have made more effort to connect their work to the ML literature, which is important for NIPS. Application: Distortion measures are used in theoretical analysis, but It is not clear how they arise in application at all. It is too costly to compute the distortion of a given embedding for assessing its quality, and doing this to choose between candidate embeddings seems unlikely in practice. If anything, distortion measures are used in designing objective functions to find a good embedding between two explicit spaces (eg. dimension reduction), and this raises computational issues (eg. how to efficiently optimize over them) which are not mentioned in this paper. The embedding results in theorems 3,4 appear to be of the theoretical kind that offer qualitative insight but have no bearing in practice. One example work in this vein is "Spectral Hashing" (Weiss, Torralba, Fergus, NIPS'08). Variants of the "spectral" distortion they use as objective (their eq.(3)) are common in many other ML works as well. I am not sure if it squarely fits into one of the distortion measures discussed here (it seems to be a form of l_2-distortion, but I did not check in detail), but it looks related at least conceptually. The experiments utilize only synthetic data, but I find it a reasonable choice in the context of the paper. In terms of clarity, The paper is well written and organized and was a pleasant read. One point I would consider making clearer is doubling dimension, which is mentioned several times but is not properly explained. The formal definition might be unsuitable for the main text, but a short explanation about intrinsic dimension could help, and some relevant citations could help more (eg. "Efficient Classification for Metric Data" (Gottlieb, Kontorovich, Krauthgamer, COLT'10)). The supplementary PDF seems to be corrupted -- I could not open it despite several attempts on different machines. From the main text I gather that it contains the proofs of the theorems and perhaps additional simulation results. My understanding is that the proofs are either relatively straightforward given the definitions (theorems 1 and 2) or mostly follow from prior work (theorems 3 and 4). The simulations included in the main file are sufficiently complete to make the point they claim to make. Overall the main text stands well on its own. I am therefore willing to disregard the unavailability of the supplementary file for the sake of this review (and in particular to tentatively believe the claimed theorems). Question: Are there direct connections between the notions of distortion? Eg. does epsilon worst-case distortion imply (say) epsilon l_p-distortion of sigma-distortion? This could help make sense of the embeddability results. === CONCLUSION === Pros: I find the systematic study of distortion measures to be interesting and valuable. The discussion and formalization of desirable distortion properties is thorough and well-thought. The formal and experimental comparison is well-established and yields interesting insights. Overall the paper makes a solid and worthwhile contribution. Cons: The motivation and relevance of this work to machine learning remains somewhat tenuous. The introduction could use a better discussion of the motivation, and relevant prior work in the machine learning literature could be better surveyed. I do think this paper is in scope for NIPS, and that the connection could be established convincingly with some effort on part of the authors. I hope they somewhat improve this point in the final version, as it will help make their work accessible and appealing to the audience in the venue they chose to target. *** Update after author response: I am somewhat surprised by the firm practicality claims in the rebuttal, and still not quite convinced. The authors write that even "simply" reading the distance matrix takes quadratic time; true, but there is rarely justification for fully reading (nor generating to begin with) the whole distance matrix, and quadratic time is largely considered prohibitive or at least undesirable for even moderate size data. Perhaps it makes sense for the setting the authors have in mind, and this ties back to the issue of better explaining the context of their work. The conclusion of my review remains the same.